# Association of Neuroimaging Data with Behavioral Variables: A Class of Multivariate Methods and Their Comparison Using Multi-Task FMRI Data

**DOI:** 10.3390/s22031224

**Published:** 2022-02-05

**Authors:** M. A. B. S. Akhonda, Yuri Levin-Schwartz, Vince D. Calhoun, Tülay Adali

**Affiliations:** 1Department of Computer Science and Electrical Engineering, University of Maryland Baltimore County, Baltimore, MD 21250, USA; adali@umbc.edu; 2Department of Environmental Medicine and Public Health, Icahn School of Medicine at Mount Sinai, Manhattan, NY 10029, USA; yuri.levsch@gmail.com; 3Tri-Institutional Center for Translational Research in Neuroimaging and Data Science (TReNDS), Georgia State University, Georgia Institute of Technology, and Emory University, Atlanta, GA 30303, USA; vcalhoun@gsu.edu

**Keywords:** data fusion, fMRI, ICA, IVA, neuroimaging, neuropsychology

## Abstract

It is becoming increasingly common to collect multiple related neuroimaging datasets either from different modalities or from different tasks and conditions. In addition, we have non-imaging data such as cognitive or behavioral variables, and it is through the association of these two sets of data—neuroimaging and non-neuroimaging—that we can understand and explain the evolution of neural and cognitive processes, and predict outcomes for intervention and treatment. Multiple methods for the joint analysis or fusion of multiple neuroimaging datasets or modalities exist; however, methods for the joint analysis of imaging and non-imaging data are still in their infancy. Current approaches for identifying brain networks related to cognitive assessments are still largely based on simple one-to-one correlation analyses and do not use the cross information available across multiple datasets. This work proposes two approaches based on independent vector analysis (IVA) to jointly analyze the imaging datasets and behavioral variables such that multivariate relationships across imaging data and behavioral features can be identified. The simulation results show that our proposed methods provide better accuracy in identifying associations across imaging and behavioral components than current approaches. With functional magnetic resonance imaging (fMRI) task data collected from 138 healthy controls and 109 patients with schizophrenia, results reveal that the central executive network (CEN) estimated in multiple datasets shows a strong correlation with the behavioral variable that measures working memory, a result that is not identified by traditional approaches. Most of the identified fMRI maps also show significant differences in activations across healthy controls and patients potentially providing a useful signature of mental disorders.

## 1. Introduction

The availability of multiple datasets that provide complementary information is becoming increasingly common in cognitive neuroimaging [1,2,3,4,5,6]. A significant challenge while analyzing such datasets is the development of fusion methods in which the datasets can fully interact with and inform each other when identifying useful features for further study [7,8,9]. Methods based on matrix and tensor decompositions use latent variable models and allow such interactions among the datasets [10,11,12]. Since these methods provide uniqueness guarantees with few assumptions, the estimated latent variables (components) are directly interpretable, that is, one can attach physical meaning to them and help explain the relationship of datasets and populations. An essential step following the extraction of such informative components that can be interpreted as putative biomarkers of neurological conditions, such as depression and schizophrenia, is associating those with neuropsychological variables (or broadly non-imaging) such as behavioral variables [13,14,15]. This is important not only for the explanation or prediction of the brain regions associated with the behavioral variables but also for the detection and identification of disease sub-types, among other tasks [16,17,18,19,20,21,22].

Independent component analysis (ICA), a popular matrix decomposition-based technique and our focus in this paper, provides an attractive foundation for fully multivariate data fusion [8,23,24]. The use of ICA and its extensions developed for the fusion of multiple datasets can help explain the underlying relationship across datasets by starting from the assumption of the independence of latent variables (components) and are successful when there is a good model match, that is, the model’s assumptions are satisfied [25,26,27,28,29]. One such extension, independent vector analysis (IVA) [23,30], generalizes ICA to multiple datasets by exploiting the available statistical information within the datasets as well as across the datasets, critical for multivariate data fusion. IVA allows the datasets to fully interact with each other in a symmetric manner by letting the datasets play a similar role in the decomposition and establish association only when it is available [23,30]. As such, IVA provides a fully multivariate approach for the analysis of multiple neuroimaging datasets [8,31].

While the fusion of imaging datasets continues to develop, the joint analysis of imaging and non-imaging datasets, a critical component to attribute meaning to neuroimaging fusion studies via links to behavior, is fairly new and has not yet received as much attention [22,32,33]. For example, identifying the association between imaging components and behavioral variables can be used not only to interpret the estimated neuroimaging results, that is, to explain the brain areas corresponding to the observed behaviors, but also to detect neuroimaging patterns that can be used to predict individuals cognitive or behavioral performance [19,20,34]. Furthermore, these imaging components and their corresponding subject co-variations can also be used to identify subtypes of disease and subject sub-groups, which is key to enabling personalized treatments [21,35,36].

Despite the potential uses, the identification of multivariate associations between imaging and behavioral datasets through a joint analysis poses multiple significant challenges. The primary source of these challenges stems from the significant difference in the nature and dimensionality of the datasets. Since the dimensionality of the behavioral datasets tends to be significantly smaller than the imaging datasets, for example, in the few hundreds compared to hundreds of thousands in imaging datasets, fusion results can be affected by this difference in dimensionality [37,38]. Furthermore, assumptions about the imaging features may not hold for the behavioral features due to differences in the nature of the data. Therefore, most of the previous work exploring the relationship between neuroimaging and behavioral datasets focus primarily on performing individual analyses on each neuroimaging dataset and conducting univariate comparisons or using one to predict another [39,40,41,42]. A few recently published methods use fusion-based methods on neuroimaging datasets and show improved performance compared with individual analysis methods [33,43,44]. However, in both the individual and fusion analysis-based approaches, simple correlation-based methods are used to identify associations between imaging and behavioral features [22,39,41,43]. Unfortunately, these pairwise analysis methods do not fully leverage the available information across datasets. One natural way to address this problem is to divide it into parts, fuse the imaging datasets first, obtain summary statistics, and then jointly analyze the behavioral variables with the summary statistics to identify multivariate associations across imaging and behavioral features, which we propose.

In this work, we focus on identifying multivariate associations between imaging and behavioral datasets and present a novel two-step approach based on IVA. We first make use of the strength of IVA to allow full interaction among the datasets to estimate joint independent components and their corresponding columns of mixing matrices, that is, subject co-variations. Subject co-variations provide an efficient way to summarize the imaging fusion results across multiple datasets, and at the same time, offer a natural way to discover associations with the behavioral variables [10]. Then in the second step, we demonstrate two unique ways to identify multivariate associations between the imaging data summary provided by the IVA and the behavioral variables, again by making use of IVA, but through two different approaches designed to discover multivariate associations across datasets. The proposed two-step approach alleviates the dimensionality, and the model mismatch issues of the imaging and the behavioral datasets by dividing the problem into parts and fusing subject co-variations with the behavioral variables. We demonstrate the performance advantages of our proposed method over existing ones first through carefully designed simulations and then using real-world data examples. With simulations, we show that our approach involving IVA improves the success rate of identifying the associated imaging and behavioral components in relation to accuracy and correlations with ground truths. We then demonstrate the successful application of our proposed scheme on real multitask functional magnetic resonance imaging (fMRI) data and behavioral variables collected from collected from 138 healthy controls and 109 patients with schizophrenia. The three tasks in this study involve an auditory oddball (AOD) task, a Sternberg item recognition paradigm (SIRP) task, and a sensorimotor (SM) task [45]. The cognitive assessments we use in this study are collected from the subjects using the Wechsler adult intelligence scale-third edition (WAIS-III) [46], Wechsler memory scale-third edition (WMS-III) [47], and Hopkins’s verbal learning test–revised (HVLT) [48]. We select these tests based on their ease of administration, good reliability of scoring, and ability to assess cognitive impairments more accurately in the schizophrenia patients [45]. We use the latter-number subtest of WAIS-III to measure the working memory, the face recognition subtest of WMS-III to measure the visual memory, and the logical memory test of WMS-III and HVLT to measure verbal memory and learning. We show that the proposed method identifies relationships across certain fMRI components and behavioral variables unidentified by the pairwise correlation technique. We identify profiles corresponding to the fMRI components that showing activation in the motor, sensory motor, dorsolateral prefrontal cortex (DLPFC) and lateral posterior parietal cortex (LPPC) is significantly correlated (p<0.05) with the behavioral variable responsible for measuring working memory. These areas have been linked with working memory deficits in schizophrenia in multiple prior reports [32,49,50], hence improving our confidence in the proposed method. The results also indicate that components that show a strong association with the behavioral variables in most cases show strong group differences between healthy controls and patients across multiple datasets, hence can be treated as candidate biomarkers of disease. The article is organized as follows. In Section 2, we introduce and discuss our new method and in Section 3, we discuss the implementation and results. We provide conclusions in the final section.

## 2. Materials and Methods

The joint decomposition of many neuroimaging datasets collected from a group of subjects is difficult. A significant issue is the high dimensionality of the datasets. Therefore, it is desirable to reduce each subject’s dataset into a feature and to obtain a lower-dimensional representation while keeping as much variability in the data as possible [51,52]. Let us consider $X_{k}\in R^{M\times T},\;k=1,2,\cdots ,K$ as a grouping of *K* neuroimaging datasets collected from *M* subjects, where the *m*th row of each dataset is formed by extracting one multivariate feature for each subject. Such reductions alleviate the high dimensionality problem and offer a natural way to discover association across these feature datasets, that is, the variations across subjects. We assume that the datasets are formed from a linear mixture of *M* independent components—referred to as sources—the noiseless ICA model for the *k*th dataset is formulated by
(1)$$x_{k}\left(t\right)=A_{k}s_{k}\left(t\right),\;k=1,2,\cdots ,K,\;t=1,2,\cdots ,T,$$
where $A_{k}\in R^{M\times M},k=1,2,\cdots ,K$ are invertible mixing matrices. Given the model, the ICA solution finds a demixing matrix $W_{k}$ such that the estimated components are statistically independent within the *k*th dataset and can be computed using $u_{k}\left(t\right)=W_{k}x_{k}\left(t\right)$. For a given set of observations $X_{k}$, the above equation can be written as $U_{k}=W_{k}X_{k}$, where $X_{k},U_{k}\in R^{M\times T}$ and $U_{k}={\left[u_{k}^{\left[1\right]},u_{k}^{\left[2\right]},\cdots ,u_{k}^{\left[M\right]}\right]}^{T}$. The columns of estimated demixing matrices, ${\hat{A}}_{k},\;k=1,2,\cdots ,K$ are referred to as subject co-variations, or profiles as they quantify the contributions of subjects to the overall mixtures $X_{k}$.

For *K* datasets, association across the datasets can be established in two different ways: either using the estimated components and/or the profiles. These associations across datasets are called *links*, which can either be hard or soft [10]. The hard link implies that the estimated components and/or profiles are identical—that is, common—across datasets, where the soft link implies the statistical dependence of components and/or profiles across datasets. If the datasets are all of the same type, dimension, and resolution, as in multiset data, we can use both components and profiles to establish links across datasets [53,54]. Examples of such multiset data are fMRI datasets from multiple subjects or tasks, or EEG data from multiple channels or frequencies [55,56,57,58]. However, if the datasets are different in type, dimension, or resolution, as in multimodal data, we can neither assume the components are identical nor statistically dependent across datasets [59,60,61]. Thus, the only possible way to link them is through their estimated profiles. Examples of multimodal data are fMRI, Electroencephalography (EEG), or structural MRI (sMRI) data from multiple subjects or tasks, or MRI and spectroscopy data from multiple subjects [62,63,64].

Now, let us consider $y_{z}\in R^{M},\;z=1,2,\cdots ,Z$ to be the collection of *Z* behavioral features collected from *M* subjects using different cognitive tests. The behavioral dataset $Y\in R^{M\times Z}$ is formed by putting all *Z* behavioral features side by side. A direct way to jointly analyze neuroimaging and behavioral features is by treating the behavioral dataset as another modality and jointly decompose it with the imaging modalities using fusion techniques [65]. However, such a direct approach is challenging due to multiple factors. An important one among those is the large dimensionality (sample) mismatch between the imaging and behavioral datasets. The contribution of the datasets to the fusion results can be significantly different due to the large difference in their sample sizes (Z≪T), which might bias the fusion results and affect the interpretability of the estimated results [37,38]. Furthermore, elements in each row of the behavioral dataset, $y_{z}^{\left[m\right]},\;z=1,2,\cdots ,Z,\;m=1,2,\cdots ,M$, can be inherently very different as they come from different behavioral experiments. Thus, the same assumptions of imaging components cannot be placed on the behavioral components while decomposing the datasets jointly. Since the imaging and behavioral datasets are inherently very different, one natural way to establish a connection between these disparate datasets is through subject co-variations or profiles. This involves dividing the problem into parts, first, decompose the imaging datasets and obtain summary statistics through profiles, and second analyze the behavioral variables with the profiles obtained in the first step to identify the associated profiles and behavioral features, which we discuss next. For the simplicity of our discussion, we refer to the estimated subject co-variations or profiles ${\hat{a}}_{k}^{\left[m\right]}$’s as imaging features and columns of behavioral dataset $y_{z}$’s as behavioral features.

### 2.1. Methods to Identify Associations between Imaging and Behavioral Features

In this section, we first introduce a general umbrella for the methods to identify associations between imaging and behavioral features.

Based on the number of variables used, associations between ${\hat{a}}_{k}^{\left[m\right]}$’s and behavioral features $y_{z}$’s can be identified in three general ways: one-to-one, one-to-many, and many-to-many association techniques, as shown in Figure 1. We define the one-to-one technique as a pairwise analysis technique where associations between ${\hat{a}}_{k}^{\left[m\right]}$’s and $y_{z}$’s are estimated in pairs. Examples of the one-to-one approach are different forms of correlation analysis. This is the most commonly used technique found in the literature when studying associations between imaging and behavioral information [41,66,67,68]. A one-to-many approach, on the other hand, identifies the association between a single behavioral feature ($y_{z}$) and all imaging features ($A_{k}$) or vice-versa, as shown in Figure 1b. This type of joint analysis is helpful for simultaneously identifying multiple imaging features associated with a single behavioral feature. Finally, the many-to-many approach, as shown in Figure 1c, takes all the available information, from both (${\hat{A}}_{k}$) and (Y), into account to identify the multivariate associations between imaging and behavioral features. The many-to-many approach facilitates the simultaneous identification of multiple brain areas associated with one or more behavioral features. Both one-to-many and many-to-many techniques use joint analysis and allow interaction among the features across imaging and behavioral datasets in a data-driven way. Hence, these approaches can identify multivariate associations and have the potential to discover relationships one cannot identify using the one-to-one approach.

To the best of our knowledge, this is the first work that aims to introduce the one-to-many and many-to-many techniques to identify the association between imaging and behavioral features in a fusion framework. A newly introduced method in [69] uses an indirect version of the one-to-many technique to estimate imaging components that share strong associations with the behavioral features. The method uses behavioral features as references to guide the decomposition of imaging datasets, thus constraining the process, potentially reducing the interpretability of the estimated imaging components. In this work, we are interested not only in accurately identifying the imaging components associated with the behavioral variables but also in the quality of the estimated components, that is, estimating components that are physically meaningful; hence we will not focus on the method introduced in [69]. Most of the other prior work which uses the one-to-one association technique use separate analyses of the individual imaging datasets, thus not allowing interaction among the datasets while estimating the imaging components [41,66,67,68]. A practical way to achieve both high quality imaging components and multivariate associations with behavioral features, as we propose in this paper, is to split the problem into parts, estimating the imaging components first through a proper fusion framework, and then identifying the imaging features associated with behavioral features using multivariate association techniques. The details about the proposed work and steps to perform are described next.

### 2.2. A Novel Two-Step Approach to Identify Multivariate Associations between Imaging and Behavioral Features Using IVA

#### 2.2.1. Step 1: Estimation of the Imaging Features

The first step of the method, as shown in Figure 2, is to use IVA to jointly decompose the imaging datasets. IVA generalizes ICA to multiple datasets such that statistical dependence among the dataset can also be taken into account and is shown to provide an attractive solution for fusion of neuroimaging datasets [31,70]. Given the problem in (Equation 1), IVA solution finds *K* demixing matrices $W_{k},\;k=1,2,\cdots ,K$ such that source components from each dataset can be estimated jointly through $u_{k}\left(t\right)=W_{k}x_{k}\left(t\right)$. The estimated components are independent within a dataset while maximally dependent on corresponding components across the datasets. This way, IVA takes the dependence among datasets into account, to obtain decompositions that fully leverage the commonalities across the datasets. This is done by modeling the source component vector (SCV), where the *m*th SCV is defined as:(2)$$\begin{array}{c} s_{m}\left(t\right)={\left[s_{1}^{\left[m\right]}\left(t\right),s_{2}^{\left[m\right]}\left(t\right),\cdots ,s_{K}^{\left[m\right]}\left(t\right)\right]}^{T}\in R^{K},\;m=1,2,\cdots ,M, \end{array}$$
that is, by concatenating the *m*th source components from each of the *K* datasets, where $s_{k}^{\left[m\right]}\in R^{T}$ is the *m*th source from the *k*th dataset. IVA thus maximizes independence across the SCVs by minimizing the mutual information [23]
(3)$$J_{\mathrm{IVA}}\left(W\right)=\sum_{m=1}^{M}\left[\sum_{k=1}^{K}H\left(u_{m}^{\left[k\right]}\right)-I\left(u_{m}\right)\right]-\sum_{k=1}^{K}\log\left|\det W^{\left[k\right]}\right|,$$
where $H\left(u_{m}^{\left[k\right]}\right)$ denotes the entropy of the *m*th source estimate for the *k*th dataset. $I\left(u_{m}\right)$ denotes the mutual information within the *m*th SCV. Here, the link across the datasets is established using a suitable multivariate probability density function (pdf) for the SCVs. Depending on the pdf used to model the SCVs, IVA can take either second or higher order statistics or both into account. In our formulation, we use IVA using multivariate Laplace distribution where both second-order and higher-order information are taken into account (IVA-L-SOS) to fuse the imaging datasets [71]. IVA-L-SOS provides a suitable model for neuroimaging datasets, especially for the fMRI datasets we use in this study. Since the source components estimated in fMRI applications correspond to brain regions with heavy-tailed distributions [72,73], an algorithm that takes both second and higher-order information into account, such as IVA-L-SOS, is preferable. In addition, the estimated subject profiles, ${\hat{a}}_{k}^{\left[m\right]}$, m=1,2,⋯,M and k=1,2,⋯,K explain each subject’s contribution to the components correlated across datasets, and hence provide a useful summary through fusion. Here, it is important to note that IVA-L-SOS provides a good match to the properties of the fMRI data, and hence is an attractive solution [71,74]. For simplicity, we do not take sample dependency into account in the rest of the article and consider only independent and identically distributed samples, thus dropping index *t* in (Equation 1).

#### 2.2.2. Step 2: Association with Behavioral Variables

In the second step, we use the estimated profile matrices, ${\hat{A}}_{k}\in R^{M\times M}$,k=1,2,⋯,K, from the first step to identify multivariate associations with the behavioral features $y_{z},\;z=1,2,\cdots ,Z$. Note that, The profiles provide an effective summary of the neuroimaging data. As shown in Figure 3, we propose two different techniques, ac-IVA and IVA-G, to identify the associations. Here, ac-IVA uses a behavioral feature at a time to constrain the imaging features so that the correlations with imaging features are maximized, thus using the one-to-many associations. In comparison, IVA-G simultaneously uses all the behavioral features and profile matrices and transforms them into a space where the correlation among all is maximized. Thus, using the many-to-many associations. Identifying the profiles associated with behavioral features will reveals the identities of the imaging components, ${\hat{s}}_{k}^{\left[m\right]}\in R^{M}$, connected to the subjects’ behavioral aspects. We start with the ac-IVA [71] and then discuss the IVA-G [75] approach to identify multivariate associations between profiles and behavioral features. Since both methods use SCV correlation matrices to identify associated profiles and behavioral features, we will use correlation instead of association in the rest of the article for simplicity.

The ac-IVA algorithm, also known as parameter-tuned constrained IVA, is a semi-supervised technique that uses reference signals to influence the IVA estimated results [71]. The reference signals can be based on any prior information. In our formulation, as shown in Figure 3a, we use behavioral features $y_{z},\;z=1,2,\cdots ,Z$ as reference signals to constrain the decomposition of the profile matrices ${\hat{A}}_{k}$,k=1,2,⋯,K. The goal here is to use the estimated $c_{k}^{\left[m\right]}$’s and their corresponding columns of $B_{k}$ to identify the ${\hat{a}}_{k}^{\left[m\right]}$’s that are maximally correlated with referenced signals. ac-IVA does that by adding a constraint $g({\hat{a}}_{k}^{\left[m\right]},y_{z})$ to the IVA cost function in (Equation 3) [71], where
(4)$$g\left({\hat{a}}_{k}^{\left[m\right]},y_{z}\right)=\rho_{m}-r\left({\hat{a}}_{k}^{\left[m\right]},y_{z}\right)\leq 0.$$

Here, r(·,·) is the function that defines the similarities between the profiles and the behavioral features, and $\rho_{m}$ is the constraint parameter, selected adaptively from a set of possible values ranges from 0.1 to 0.99. We use Pearson’s correlation as the similarity function r(·,·). Thus, the absolute value of r(·,·) is bounded by 0 to 1 limiting the $\rho_{m}\leq 1$. A higher the value of $\rho_{m}$ imposes a harder constraint on the decomposition, while a lower value reduces the power of the constraint. Thus, $\rho_{m}$ values indicate the strength of correlation of a behavioral feature over *M* subject profiles. This, along with the elements in the columns of $B_{k}$ in Figure 3a, can identify the profiles that are correlated with the *z*th behavioral feature. Indices of the correlated profiles reveal the identities of the corresponding imaging components that relate to the *z*th behavioral feature. We use a single behavioral feature at a time as a reference signal, thus using the one-to-many association technique, and repeat the experiment *Z* times for all *Z* behavioral features.

IVA-G, on the other hand, uses all the behavioral features and the profile matrices simultaneously, thus using the many-to-many association technique, to identify the correlated imaging and behavioral features. Generative model of IVA-G is shown in Figure 3b. IVA-G uses multivariate Gaussian distribution to model the SCVs, thus taking second-order information of the datasets into account [23,75]. Given ${\hat{A}}_{k}\in R^{M\times M}$,k=1,2,⋯,K and $Y\in R^{M\times Z}$, and M=Z, IVA-G jointly transforms the datasets into a space where the correlation between $c_{k}^{\left[m\right]}$’s, where k=1,2,⋯,K,K+1, are maximized. The elements of the corresponding columns of the $B_{k}$’s reveal the identities of the profiles in ${\hat{A}}_{k}$; thus, identities of the sources in $U_{k}$ that are strongly correlated to the behavioral features in Y. In the case of M≠Z, we generate M−Z signals using standard Gaussian distribution and stack them with the behavioral matrix if M>Z or with profile matrices if M<Z to ensure equal dimensions of the datasets. We call the added standard Gaussian signals as *padding* signals and use those to make dimension of the datasets similar. It is important to note here that these additional signals do not change the results’ outcome but rather allow joint decomposition by alleviating the dimensionality mismatch issue in profile and behavioral matrices. Since IVA-G uses all the datasets simultaneously, it utilizes more information and provides a more accurate estimation of the identities of correlated profiles and behavioral features.

The proposed two-step method to jointly analyze imaging and behavioral datasets alleviates the dimensionality mismatch issue of the datasets. Its disjoint nature also gives us the flexibility to choose appropriate algorithms to decompose imaging datasets in step one and fuse imaging and behavioral features in step two. In the first step, where the dimensionality of the imaging datasets is large, using algorithms that take all order statistics into account, as in IVA-L-SOS, ensures proper estimation of interpretable brain maps and useful summary statistics across populations. Conversely, in the second step, where the number of samples is the number of subjects, and thus small, using flexible data-driven algorithms allows efficient estimation of the correlation information across the imaging and behavioral features. As we show in the next section, through simulations and real data analysis, the proposed approach can estimate correlated features more accurately and identify underlying relationships across imaging and behavioral datasets that cannot be identified by traditional techniques.

## 3. Results

### 3.1. Simulation Setup and Results

We perform two different simulation experiments. In the first set of experiments, we compare the performance of ICA and its three extensions for fusion, jICA [25], DS-ICA [76], and IVA [71], to estimate and identify the neuroimaging components associated with BVs using a simple one-to-one correlation technique. We show that fusion analysis provides a better estimation of correlated components than the separate analyses of individual datasets, and among the fusion methods, IVA provides better estimation performance. In the second set of experiments, we use IVA to estimate the neuroimaging components and compare the performance of one-to-one, one-to-many, and many-to-many techniques to identify the correlation between neuroimaging and behavioral features. We compare the performance of correlation, regression, ac-IVA, and IVA-G to accurately identify the correlated components. We show that ac-IVA and IVA-G identify correlated features more accurately than correlation and regression analysis techniques.

We generate simulated examples for K=3 datasets. For each dataset, we generate N=10 sources from the Laplacian distribution with T= 10,000 independent and identically distributed (i.i.d.) samples. We select the Laplacian distribution because it is a good match to fMRI data [71,77]. We introduce correlation to three sources across the datasets with correlation values randomly selected from 0.3 to 0.9. These are the source that are correlated or common across datasets, and the seven other components are distinct. These sources are then linearly mixed with mixing matrices $A\in R^{M\times 10}$ with elements from a standard Gaussian distribution N(0,1), where *M* is the number of subjects from two groups—a healthy control group and a patient group—each with M/2 samples (subjects). To simulate group differences in the subjects, a step-type response with step height *h* is added to the first column of each mixing matrix or profile of all three datasets. Thus, we have profiles $a_{k}^{\left[1\right]}\left(m\right)=v_{k}^{\left[1\right]}\left(m\right)+1.5u\left(m\right),\;m=1,2,\cdots ,M$ for k=1,2,3. Here, *M* is the even number of subjects, $v_{i}\left(m\right)$’s are generated using standard Gaussian distribution and u(m) is the step response with $u\left(m\right)=\left\{\begin{array}{cc} 1, & m\leq M/2\\ 0, & m>M/2 \end{array}\right.$. The standard deviations of other profiles are adjusted such that they match the standard deviation of the discriminant profiles for each case we consider. We refer to the components corresponding to these discriminative profiles as discriminative components. We use an order of 10 for dimensionality reduction resulting in $X_{k}\in R^{10\times 10,000}$ for k=1,2,3. We generate B=4 behavioral features from standard Gaussian distribution N(0,1) with *M* i.i.d. samples resulting in $Y\in R^{4\times M}$. We choose to generate four behavioral features to be consistent with the real data used in this study, which we will discuss later in the paper. A step-type response with step height *h*, similar to the one added to the discriminative profiles, is added to the first and fourth behavioral features. The purpose of this is to introduce correlation between behavioral features and discriminative profiles. We evaluate the performance of each method to identify the correlated profiles and behavioral features by either changing the number of subjects *M* or the step-height *h* used to distinguish two subject groups. In the first case, the number of subjects is changed from 50 to 500 with step-height fixed to 2, while in the second case, the step-height is changed from 0.5 to 3—resulting in correlation values in the range [0.3, 0.9]—with subject number fixed to 300.

For the first set of experiments, we evaluate the performance of individual ICAs, jICA, DS-ICA and IVA to estimate and accurately identify the correlated profiles and behavioral features. We estimate one-to-one correlations between the estimated profiles and the behavioral features and use p<0.05 as a threshold to identify the correlated ones. We use the accuracy of the methods to correctly identify the target discriminative profiles, and average correlation between the true and the estimated discriminative components to evaluate each method’s performance. We estimate the accuracy using:(5)$$\mathrm{Accuracy}=\frac{TP+TN}{N}\times 100\%,$$
where TP’s are the true positive number of profiles correlated with the behavioral features and TN’s are the true negative number of profiles uncorrelated to the behavioral features. We use the same ICA by entropy bound minimization (ICA-EBM) [78] algorithm for individual ICAs, jICA, and DS-ICA, while using the IVA-L-SOS algorithm for IVA.

Figure 4 shows the performance comparison of the methods for (a) different numbers of subjects and (b) different step-heights. In Figure 4a,b, the accuracy of identifying the correlated profiles and the component correlation improve for all methods as we increase the subject’s number and the step-heights. The fusion methods—jICA, DS-ICA, and IVA—provide better performance than individual ICAs since they use complimentary information available across datasets. Among the fusion methods, IVA and DS-ICA provide better performance than jICA likely because these two methods are less constrained and allow better interaction among the datasets. In addition, IVA outperforms DS-ICA when the subject’s numbers are lower, or step-heights are small, indicating better flexibility of the IVA algorithm.

For the second set of experiments, we only use IVA to estimate the components and their corresponding profiles since it provides better performance than the other methods. Here, we evaluate the performance of four association identification techniques—correlation, regression, ac-IVA and IVA-G—to identify the profiles correlated with behavioral features. We use accuracy defined in (Equation 5) to evaluate the performance of these techniques.

Figure 5 compares the performance of the methods for (a) different numbers of subjects and (b) different step-heights. Th accuracy of identifying the target profiles improves for all techniques as we increase the subject’s number and the step-heights. IVA-G, ac-IVA, and regression analysis, which use either the many-to-many or the one-to-many analysis schemes, provide a better identification performance compared with the simple one-to-one correlation analysis technique. The two proposed techniques, IVA-G and ac-IVA, outperform regression analysis, and IVA-G provides a slightly better performance compared with ac-IVA to identify the correlated profiles and behavioral features. To summarize, IVA provides a better estimation performance among the ICA-based fusion methods in the first set of experiments. Meanwhile, IVA-G provides a better identification performance than correlation, regression, and ac-IVA in the second set of experiments.

### 3.2. FMRI Data and Extracted Features

The multitask fMRI datasets and the behavioral features used in this study are from the Mind Research Network Clinical Imaging Consortium Collection [45]. These datasets and the behavioral features are publicly available at https://coins.trendscenter.org/ (accessed on 23 December 2022). The fMRI datasets were collected from 247 subjects, 138 healthy individuals and 109 schizophrenia patients, while performing auditory oddball (AOD), Sternberg item recognition paradigm (SIRP), and sensory-motor (SM) tasks. In addition, the behavioral scores were collected from the same subjects using the Wechsler adult intelligence scale (WAIS), the Wechsler memory scale (WMS) and the Hopkins verbal learning test (HVLT). The imaging features are generated using the statistical parametric mapping (SPM) toolbox [79]. For each task, the voxels from each subject are analyzed utilizing simple linear regression using the SPM toolbox. Here, the regressors are created by convolving SPM’s hemodynamic response function (HRF) with desired predictors. The resultant regression coefficient maps are then used as imaging features for each subject and task. Details about the fMRI datasets, extracted features and behavioral tests are discussed below.

#### 3.2.1. Auditory Oddball Task (AOD)

During the AOD task, subjects were required to listen to three different types of auditory stimuli—standard, novel and target—coming in a pseudo-random order and were required to press a button only when they heard the target stimuli [80]. The standard stimuli were frequent 1 kHz tones with a probability of occurrence of 0.82. The novel stimuli were infrequent computer-generated complex sounds with a probability of occurrence of 0.09. Finally, the target stimuli were infrequent 1.2 kHz tone with the same probability of occurrence as novel tones, except here, the subjects were required to press a button [45]. A regressor was created to model the target-related stimuli as a delta function convolved with the default SPM HRF [81] and subject averaged contrast images of target tones were used as the feature for this task.

#### 3.2.2. Sternberg Item Recognition Paradigm Task (SIRP)

During this visual task, the subjects were asked to memorize a set of 1, 3, or 5 integer digits randomly selected from integers 0 to 9. The task paradigm (each WM block) lasted for a total of 46 s, including 1.5 s of learning (prompt condition), 0.5 s of a blank screen, 6 s of encoding (encode condition), and finally, 38 s of probing (probe condition). In the probing session, subjects were shown a sequence of integers and were required to press a button whenever a digit from the memorized set arrived [45]. Six 46 s WM blocks consisting of the prompt-encode-probe conditions were used in each scan. For this task, a regressor was created by convolving a three-digit probe response block with default SPM HRF [81], and the average map was used as the feature for this task.

#### 3.2.3. Sensory Motor Task (SM)

The SM task involved subjects listening to a series of 16 different auditory tones [45]. Each tone lasted for 200 ms and was within the frequency range of 236 Hz to 1318 Hz. The tones were arranged in such a way that the first tone was set at the lowest pitch and each subsequent tone was at a higher pitch than the previous tone until the highest was reached. The order of the tones was reversed after the 16th tone. The subjects were required to press a button for each tonal change. Each run consisted of 15 increase-and-decrease blocks, alternated with 15 fixation blocks, with each block lasting for 16 s. For this task, a regressor was created by convolving the entire increase-and-decrease block with SPM HRF, and an average map was used as the feature for this task.

#### 3.2.4. Behavioral Features

This study uses cognitive measures collected from the subjects using WAIS-III, WMS-III, and HVLT tests. The letter-number sequence subtest of WAIS-III was used to evaluate the working memory of the subjects. Visual memory of the subjects was evaluated using the face recognition subtest of WMS-III. Finally, verbal memory and learning of the subjects were evaluated using the logical memory test subset of WMS-III and HVLT. We use letter-number total raw score in WAIS-III, recall total score and recognition total scores in WMS-III, and total raw in HVLT as behavioral features in our analysis. In total, we have four behavioral features from 247 subjects. We perform a two-sample *t*-test on each feature. As shown in Figure 6, all four behavioral features show substantial group differences (p<0.05) between healthy controls and schizophrenia patients, with patients reporting lower scores (more cognitive deficits) than healthy controls. It is important to note that temporal synchronization of fMRI data and behavioral variables is not needed since the task-related associations are considered through the initial multivariate feature extraction step, and behavioral associations are at the group level.

### 3.3. Order and Algorithm Selection

For all three tasks, extracted features from all 247 subjects are concatenated vertically, healthy controls and then patients, to form the feature datasets each with T= 48,546 number of voxels. The dimension of the signal subspace is then estimated for each task feature dataset using an entropy-based method proposed in [82]. Determining the signal subspace order is critical for fMRI data analysis to avoid over-fitting due to the high noise level of medical imaging data. We use an order N=25 for all three datasets resulting $X_{k}\in R^{25\times 48,546},\;k=1,2,3$. A practical way to test the stability of the estimated order is to check the stability of the estimated components for different orders around that number [31]. We check the performance of the methods for a set of orders 15,20,25,30,35, besides the estimated order of 25. The results are pretty similar in terms of activation areas for orders 20 and 30, whereas the stability of components started to change, either getting merged or split into two, for orders 15 and 35. Since there is no ground truth, we select the final order using the guidance of the selection methods and the quality and the stability of the estimated results [31,76].

We use IVA-L-SOS in the first step to decompose the task fMRI feature datasets and estimate interpretable components and their corresponding subject profiles. Since IVA-L-SOS is an iterative algorithm, it is crucial to run the algorithm multiple runs and evaluate the consistency of the estimated results to select the most consistent result. To accomplish this we run the algorithm multiple times and using cross intersymbol interference (Cross-ISI) [83] to select the most consistent run, leading to a reproducible decomposition. We use the ac-IVA [71] and regular IVA-G [75] to identify the associated profiles and behavioral features.

### 3.4. FMRI Data Results and Discussion

Since the total number of estimated components across three datasets is high (75), we visually check all the estimated components and use only the interpretable ones, that is, showing higher activation in physically meaningful brain areas and are hence useful. We select a total of 21, that is, seven components per dataset for the second step and use their corresponding profiles to estimate correlations with four behavioral features. These selected components are then thresholded at Z=2.7 and shown in Figure 7. A two-sample *t*-test is used on the profiles to test for significant group differences between healthy subjects and patients. The components corresponding to those profiles that pass the significance tests (p<0.05) are referred to as *discriminative* components or putative *biomarkers* of disease. The colormap of the components is adjusted so that the colors red, orange and yellow indicate higher activation in healthy controls, and blue means higher activation in patients. Overall, the components in Figure 7 show higher activation in the angular gyrus (AG), visual, and motor cortex areas in patients with schizophrenia while showing higher activation in default mode network (DMN), auditory cortex, and frontoparietal (FP) regions in healthy controls. The FP component is estimated in two separate components, right and left frontoparietal (RFP and LFP). Both RFP and LFP contain the dorsolateral prefrontal cortex (DLPFC) and lateral posterior parietal cortex (LPPC), which are part of the central executive network (CEN). In Figure 7, components differentiating between healthy subjects and patients (p<0.05) show higher activations in DMN, CEN, motor, and auditory cortex in AOD and SM tasks; and visual, DMN, and AG regions in the SIRP task. All these areas are known to be associated with schizophrenia [49,50,84,85,86,87,88] and therefore indicate meaningful decomposition results.

Next, we use the profiles corresponding to these components and use ac-IVA and IVA-G to identify the profiles correlated with the behavioral features. We use one behavioral feature at a time to constrain ac-IVA decomposition to identify the associated components. After ac-IVA transformation, we identify that the behavioral features show significant correlations (p<0.05) with only one SCV with the strongest correlation structure. Since most of the profiles across datasets have step-type characteristics, meaning they show significant group differences between healthy controls and patients, ac-IVA is expected to estimate an SCV that summarizes that property of the profiles. We show the ac-IVA results in Figure 8. By analyzing the columns of the mixing matrices corresponding to the SCV with the strongest correlation structure, we identify that WAIS and WMS2 show significant correlations with components from the AOD dataset, and WMS1 shows a significant correlation with the components from the SIRP dataset. WAIS and WMS2 correlate with the profiles corresponding to DMN, right and left FPs, motor, and auditory cortex components in the AOD dataset, and WMS1 correlates with the profiles corresponding to the visual cortex and RFP components in the SIRP dataset. On the other hand, IVA-G uses all the behavioral features together and uses a single decomposition to analyze behavioral features and the profile matrices jointly. Since we have four behavioral features and seven profiles in each dataset, we generate three *padding* signals and stack them with the behavioral features to have similar dimensions as profile matrices. After the IVA-G transformation, we identify the SCVs within which the behavioral components show strong correlations with the imaging components. We show the components estimated by IVA-G as correlated with behavioral features in Figure 9. Again, by analyzing the columns of mixing matrices corresponding to these SCVs, we identify that WAIS and WMS1 show correlations to the profiles corresponding to the motor cortex, RFP, and DMN components in the AOD dataset; DMN, visual cortex, and AG components in SIRP dataset; and DMN, visual, and auditory cortex components in SM dataset. These results agree with the results estimated by the ac-IVA method, thus increasing our confidence about the estimated relationships between imaging components and behavioral variables.

In both ac-IVA and IVA-G results, the two behavioral features WAIS and WMS1 show significant correlation with the neuroimaging components. WAIS, which measures verbal working memory through the letter-number sequence subtest, correlates with the profiles corresponding to the motor cortex and the FP components across task datasets. The FP components, especially DLPFC and LPPC, are related to the central executive network of the brain and responsible for the retrieval of short-term memory, self-awareness, and language-based decoding [89]. From Figure 7, Figure 8 and Figure 9, DLPFC and LPPC show higher activation in healthy subjects and lower activation in schizophrenia patients. These areas have been linked with working memory deficits in schizophrenia in multiple reports [32,49,50]. WMS1, which measures visual memory using a face recognition subtest, shows an association with the visual cortex, DMN, and AG components across task datasets. These areas of the brain are responsible for vision, visual perception, memory retrieval, attention, spatial cognition, and reasoning, among other tasks and are found to be associated with schizophrenia in many prior studies [90,91,92,93,94]. These components explain the areas of the brain associated with the behavioral features and could be used as potential predictors for cognitive deficits in schizophrenia patients. Since the behavioral features are a strong indicator of an individual’s cognitive functions, the features or the neuroimaging components correlated to these features, separately or together, can be used to identify subject groups with heterogeneous cognitive profiles. Furthermore, components correlated to WAIS and WMS1 also show strong group differences between healthy controls and patients and hence can be used as putative biomarkers of disease.

Figure 10 shows the subject profiles estimated as correlated with behavioral features using the one-to-one correlation technique. Behavioral features significantly correlate with the DMN, LFP, motor, and auditory cortex components in the AOD dataset; visual cortex and AG components in the SIRP dataset; and motor and auditory cortex components in the SM dataset. Among the behavioral features, WAIS shows correlations with the largest number of subject profiles across datasets. Overall, these results agree with the results estimated by the ac-IVA and IVA-G methods, thus increasing our confidence in the proposed methods. However, behavioral features are correlated with more subject profiles from the AOD task dataset than the SIRP and SM task dataset. Especially, behavioral features are showing significant correlation, after a Bonferroni correction, with only one component in the SM dataset. On the other hand, ac-IVA and IVA-G highlight more components in the SM dataset correlated with behavioral features. A possible explanation for this is the fact that IVA incorporates common information across the datasets that helped ac-IVA and IVA-G to identify more correlated components in the SM dataset. However, IVA will be able to take advantage of common information across datasets only if it is available, and in the absence of it, will reduce to disjoint or one-to-one analyses of individual datasets. Though we use IVA in this work, other multivariate techniques, such as canonical correlation analysis (CCA) and multiset-CCA (MCCA), can also be used instead of IVA.

## 4. Conclusions

In this work, we present a novel two-step method to estimate and identify multivariate associations between imaging and behavioral features. In doing so, we also introduce a general framework for identifying correlated imaging and behavioral features. Finally, through simulation studies and real-world data, we show the performance advantage of the proposed method over existing ones. With simulation studies, we first show that fusion analysis provides better accuracy in identifying the correlated components than performing separate analyses on individual datasets, as typically done. Then, we show that joint analysis of the imaging and behavioral features using ac-IVA and IVA-G improves accuracy in identifying the underlying relationship between imaging and non-imaging datasets than traditional one-to-one correlation techniques. With real fMRI data collected from healthy controls and patients with schizophrenia, we show that both ac-IVA and IVA-G provide a complete picture of associations between the imaging and behavioral features compared with the existing approach. The results indicate that imaging components show higher activation in the motor cortex, FP, auditory and visual cortex, showing higher correlations with the behavioral variable responsible for measuring working memory. These areas are found to be associated with working memory deficits in many prior studies and hence can be used as predictors for cognitive impairments for schizophrenia patients.

## Figures and Tables

**Figure 1 sensors-22-01224-f001:**
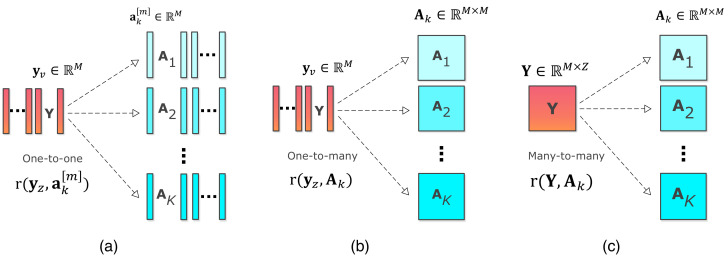
Three different ways to identify associations between imaging and behavioral features using (**a**) one-to-one, (**b**) one-to-many, and (**c**) many-to-many association techniques. Here, r(·,·) is the function that denotes the association between the profiles and behavioral features.

**Figure 2 sensors-22-01224-f002:**
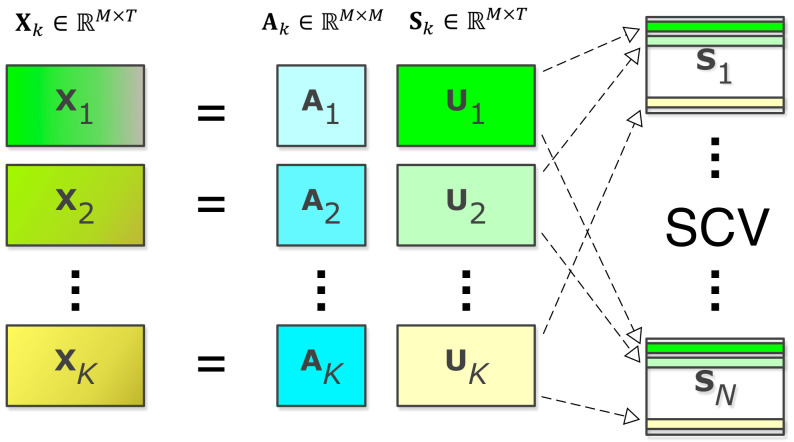
Step 1: IVA step for the fusion of neuroimaging datasets. IVA estimates interpretable components independent within the datasets while maximally correlated with the same index components across the datasets. Subject profiles corresponding to these components represent each subject’s contribution to the components and can be used to identify multivariate associations with behavioral features.

**Figure 3 sensors-22-01224-f003:**
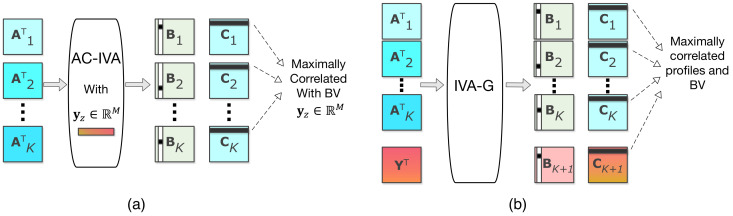
Step 2: Two different techniques to identify the association across imaging and behavioral features. Here, (**a**) represents the ac-IVA technique where a single behavioral feature is used to constrain imaging features (subjects profiles), and (**b**) represents the IVA-G approach where all the behavioral features are taken into account to identify the correlated imaging and behavioral features. In both cases, “black dot”s in the $B_{k}$ matrices represent the identities or indices of the correlated profiles or behavioral features in ${\hat{A}}_{k}^{T}$ and $Y^{T}$, which, in turn, represent indices of the components in $U_{k}$’s associated with behavioral features.

**Figure 4 sensors-22-01224-f004:**
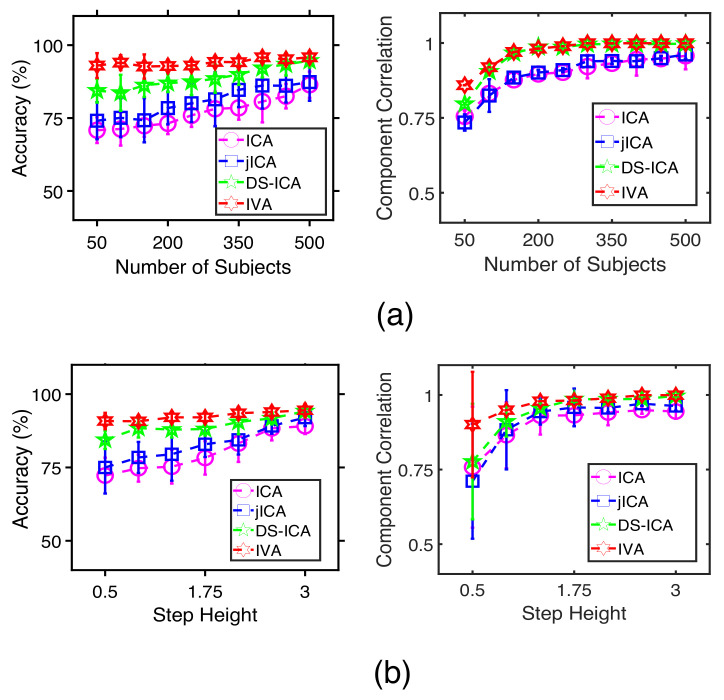
Estimation performance of individual ICAs, jICA, DS-ICA, and IVA for different (**a**) number of subject and (**b**) Step-heights. Here, correlated profiles and behavioral features are identified using the one-to-one correlation analysis technique. All the results are averaged over 100 independent runs.

**Figure 5 sensors-22-01224-f005:**
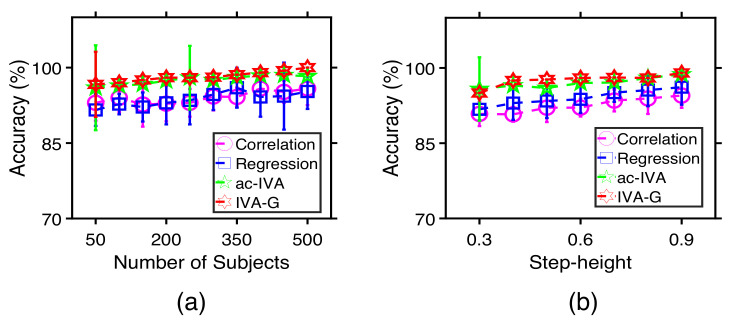
Accuracy performance of correlation, regression, ac-IVA, and IVA-G to identify the profiles correlated with the behavioral features for different (**a**) numbers of subjects and (**b**) step-heights. The components and their corresponding profiles are estimated using IVA. Performances of all the methods improve with the number of subjects and step-heights. All the results are averaged over 100 independent runs.

**Figure 6 sensors-22-01224-f006:**
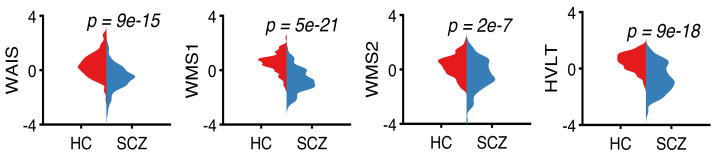
The four behavioral features used in this study are collected using the letter-number sequence subtest of WAIS-III (WAIS), the face recognition subtest of WMS-III (WMS1), and the logical memory test of WMS-III (WMS2) and HVLT (HVLT). The behavioral feature WAIS measures the working memory, WMS1 measures the visual memory, and WMS2 and HVLT measure the verbal memory and learning. All four features show strong group differences (p<0.05) between healthy controls and patients with schizophrenia.

**Figure 7 sensors-22-01224-f007:**
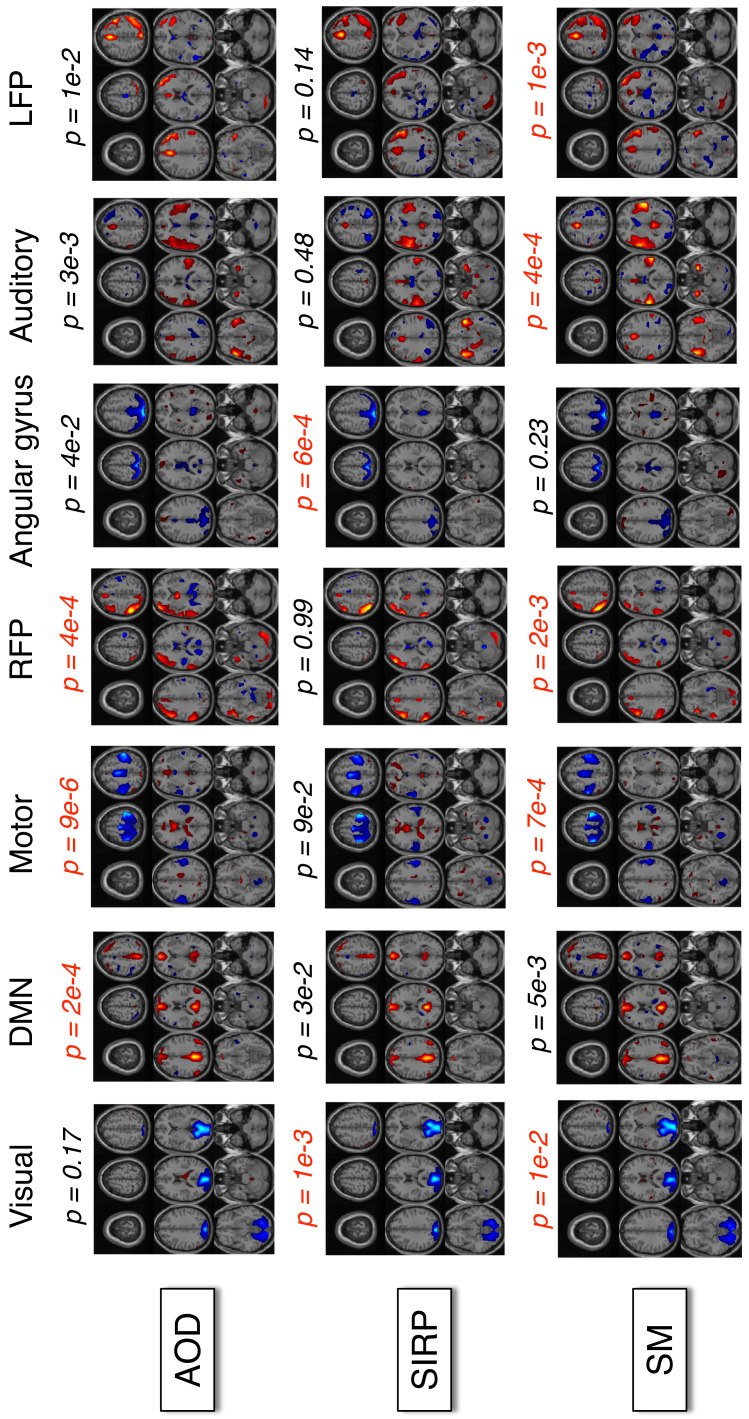
FMRI components estimated by the IVA-L-SOS algorithm. Here, the colormap of the components is adjusted so that the colors red, orange and yellow mean higher relative activation in controls and blue means higher relative activation in patients. The estimated components show significant group differences even after strict Bonferroni correction ((p×25)≤0.05) in DMN, FP, motor, and auditory cortex areas for AOD and SM tasks; and visual cortex and AG areas for the SIRP task.

**Figure 8 sensors-22-01224-f008:**
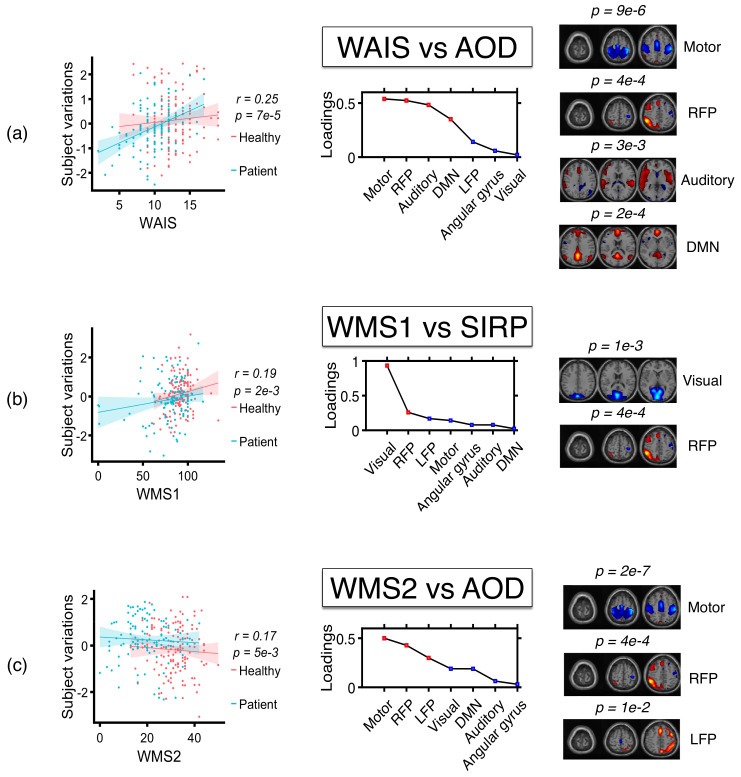
Behavioral features and subject profiles identified as correlated by ac-IVA. Here, (**a**) WAIS is correlated with the profiles associated to motor, RFP, auditory, and DMN components in the AOD task; (**b**) WMS1 is correlated with the profiles associated with visual and RFP components in the SIRP task; and finally, (**c**) WMS2 is correlated with the profiles associated to motor, and FP components in the SM task.

**Figure 9 sensors-22-01224-f009:**
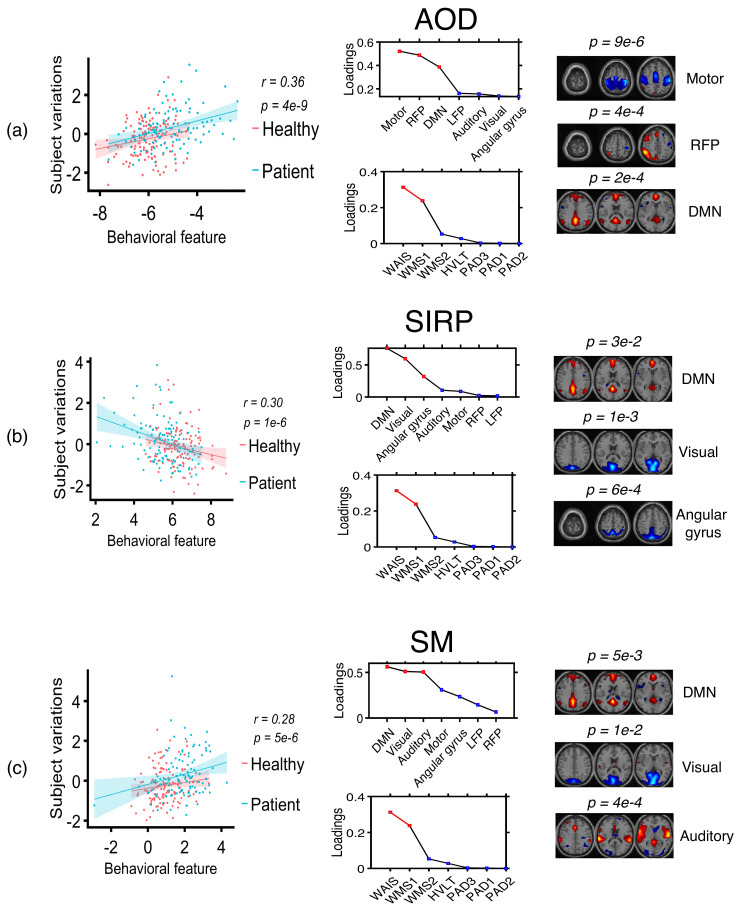
Behavioral features and subject profiles identified as correlated by IVA-G. Here, WAIS and WMS1 are correlated with the profiles associated to (**a**) motor cortex, RFP, and DMN components in AOD task; (**b**) DMN, visual cortex, and AG components in SIRP task; and (**c**) DMN, visual, and auditory cortex components in SM task.

**Figure 10 sensors-22-01224-f010:**
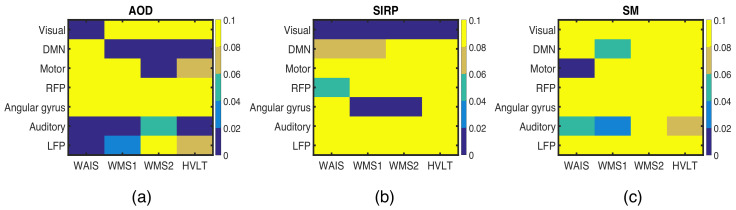
Behavioral features and neuroimaging components identified as correlated using one-to-one Pearson correlation technique across (**a**) AOD, (**b**) SIRP, and (**c**) SM task datasets. Here, we are showing *p*-values of the correlation and use p<0.05 as a threshold to identify the significantly correlated behavioral and neuroimaging features.

## Data Availability

The datasets used in this study are publicly available at https://coins.trendscenter.org/ (accessed on 23 December 2022). The codes are available at https://mlsp.umbc.edu/resources.html (accessed on 23 December 2022).

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
