# Peer review of "Association of Neuroimaging Data with Behavioral Variables: A Class of Multivariate Methods and Their Comparison Using Multi-Task FMRI Data"

_sensors, 2022, doi:10.3390/s22031224_

Round 1

Reviewer 1 Report

The manuscript submitted is on the development of a promising new technique on neuroimaging data and behavioral approaches by means of sophisticated multivariate analysis based on functional MRI images.

This is the first article using this approach that I had the opportunity to analyze. This has two different strengths or weaknesses that are connected, its innovative nature may be disappointing and may be misleading in evaluating the work, on the other hand, this may also limit the tools and pertinent previous work to hook on solid experience and evidence.

I personally was very interested when reading the work and the found the approach wortht and useful.

The authors have very adequately tackled the high dimensionality of the data of interest. Simplification and reduction of data will inevitably lead to loos of signal. I found the independent component analysis interesting ad useful. The multivariate two steps approach is also of interest.

The “synchronization” and I mean temporal synchronization of the neurophsychometric data and the FMRI data is not evident to me in the manuscript but, I can guess that Independent vector analysis doesn’t really need such a temporal synchronization.

I consider the article very interesting and wish more applications of this new method and approach that is promising and will for sure help patients in a very “personalized” way.

Thank you for giving me the opportunity of analyzing such an interesting manuscript.

Author Response

We thank the reviewer for the comment. Temporal synchronization of fMRI data and behavioral features is not needed here as task-related associations are considered through the initial regression model, and behavioral associations are at the group level. To elaborate, the single-subject fMRI features that we used are task-related and extracted using an initial general linear model (GLM) with carefully designed task-related regressors. Detail about the steps, properties of each task, and description of the corresponding task-related regressors are given in section 3.2. On the other hand, behavioral features were collected from each subject through certain standard cognitive tests. To identify the multivariate associations between behavioral and fMRI features, we first fuse the fMRI task datasets, estimate summary information of the fusion in terms of profiles, and finally estimate the association between behavioral features and profiles. To make the point clearer, we add the below sentence at section 3.2.4 in the paper:            

It is important to note that temporal synchronization of fMRI data and behavioral variables is not needed since the task-related associations are considered through the initial multivariate feature extraction step, and behavioral associations are at the group level.”

Reviewer 2 Report

This manuscript describes new methods for the joint analysis of imaging and non-imaging data obtained from a cohort of patients with schizophrenia. The neuroimaging dataset was collected from healthy subjects and patients while performing a multitasking functional magnetic resonance imaging (fMRI), which involved an auditory oddball task, a Sternberg item recognition paradigm task, and a sensorimotor task. Non-imaging data consist of behavioral variables assessed using the Wechsler adult intelligence scale-third edition, the Wechsler memory scale-third edition, and the revised Hopkins's verbal learning test. To date, simple one-to-one correlation analysis has been performed in most studies in this area, and the cross-information available across multiple datasets has not been widely used. This study proposes a two-approach method based on independent vector analysis (IVA) to jointly analyze imaging datasets and behavioral variables to identify multivariate relationships between imaging data and behavioral characteristics. The novelty introduced by this manuscript is the attempt to introduce one-to-many and many-to-many techniques to identify the association between imaging and behavioral characteristics. For this purpose, the methods are described in detail, the results and conclusions obtained are consistent. The figures are appropriate and well done. However, for figure 7 I suggest some changes to make the colormap obtained with fMRI more visible, perhaps by increasing the size of the figure or placing it horizontally.

Finally, I recommend minor spelling check: line 77 “analysis” instead of “analyses”; line 567 “associations” instead of “asociations”.

Author Response

We thank the reviewer for his helpful suggestions and insightful comments. Based on his suggestion, We make figure 7 horizontal to increase the size of the maps. We also carefully check the whole document for English use.